# AC-VAE: Learning Semantic Representation with VAE for Adaptive Clustering

## Abstract

Unsupervised representation learning is essential in the field of machine learning, and accurate neighbor clusters of representation show great potential to support unsupervised image classification. This paper proposes a VAE (Variational Autoencoder) based network and a clustering method to achieve adaptive neighbor clustering to support the self-supervised classification. The proposed network encodes the image into the representation with boundary information, and the proposed cluster method takes advantage of the boundary information to deliver adaptive neighbor cluster results. Experimental evaluations show that the proposed method outperforms state-of-the-art representation learning methods in terms of neighbor clustering accuracy. Particularly, AC-VAE achieves 95% and 82% accuracy on CIFAR10 dataset when the average neighbor cluster sizes are 10 and 100. Furthermore, the neighbor cluster results are found converge within the clustering range ($\alpha \leq 2$), and the converged neighbor clusters are used to support the self-supervised classification. The proposed method delivers classification results that are competitive with the state-of-the-art and reduces the super parameter $k$ in KNN (K-nearest neighbor), which is often used in self-supervised classification.

## 1 Introduction

Unsupervised representation learning is a long-standing interest in the field of machine learning (Peng et al., 2016a; Chen et al., 2016; 2018; Deng et al., 2019; Peng et al., 2016b), which offers a promising way to scale-up the usable data amount for the current artificial intelligence methods without the requirement for human annotation by leveraging on the vast amount of unlabeled data (Chen et al., 2020b;a). Recent works (Chen et al., 2020b;a; He et al., 2020) advocate to structure the unsupervised representation learning at the pre-training stage and then apply semi-supervised or self-supervised techniques on the learned representations in the fine-tuning stage. So the representation learning acts as a feature extractor, which extracts semantic features from the image, and well-extracted features should lead to excellent classification performance (He et al., 2020). Moreover, representation learning assigns close vectors to images with similar semantic meanings, thus making it possible to cluster the same meaning images together (Xie et al., 2016; Van Gansbeke et al., 2020). When no label is available, unsupervised or self-supervised classification methods rely on the neighbor clustering to provide the supervisory signal to guide the self-supervised fine-tuning process (Van Gansbeke et al., 2020; Xie et al., 2016). In this scenario, accurately clustering neighbors among representations is crucial for the followed classification fine-tuning.

In many of the prior unsupervised methods (Van Gansbeke et al., 2020; Xie et al., 2016), the neighbor clustering process is performed by KNN (k-nearest neighbor) based methods. However, KNN based methods introduce $k$ as a super parameter, which needs to be fine-tuned regarding different datasets. In an unsupervised setup, selecting a suitable $k$ without any annotation or prior knowledge is not straightforward. Therefore it is desirable to have a neighbor clustering process that automatically adapts to different datasets, thus eliminating the need for pre-selecting the super parameter $k$.

To achieve adaptive neighbors clustering, the proposed method tries to encode the image representation into the multivariate normal distribution, as the multivariate normal distribution provides distance information, such as z-score, which can naturally adapt to different datasets without the help of any additional mechanism. Prior works (Kingma & Welling, 2013; Higgins et al., 2016; Burgess et al., 2018) showed VAE's ability to encode images into multivariate normal distributions; nonethe-

less, these works struggled to extract high-level semantic features, as most of them were trained by image recovery tasks, which encourages the network to focus on the low-level imagery features. Consequently, the extracted low-level features cannot be utilized in the unsupervised classification method, which needs semantic features to function.

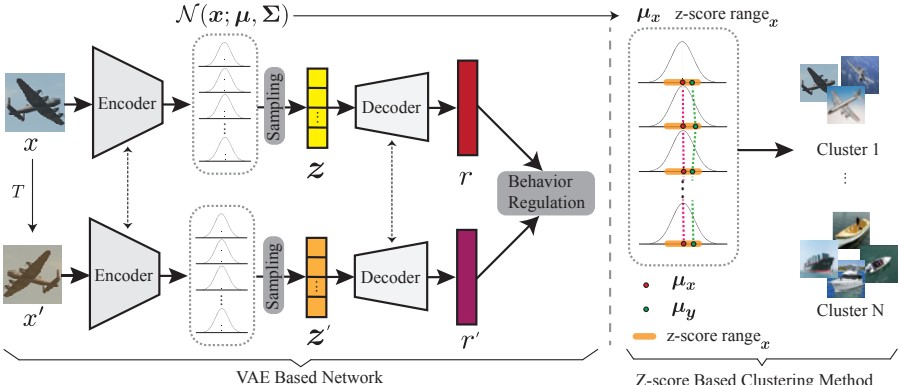

Figure 1: The proposed clustering method includes a VAE based network and z-score based cluster methods. The VAE based network encodes the image into the multivariate normal distribution, and the z-score based clustering method takes advantage of the distribution's boundary information.

To provide VAE with the ability to extract the high-level semantic features, as well as to utilize its strength to produce adaptive clusters, this paper proposes a framework, AC-VAE, including a VAE based network and a z-score based clustering method, as shown in Figure 1. The VAE based network encodes the image into the multivariate normal distribution $\mathcal{N}(\boldsymbol{\mu}, \boldsymbol{\Sigma})$, The distribution's mean $\boldsymbol{\mu}$ is taken as the representation; meanwhile, its z-score provides the boundary information that can naturally adapt to different datasets. The proposed clustering method takes advantage of the boundary information to achieve adaptive neighbor clustering. The proposed framework's efficacy is evaluated on CIFAR10, CIFAR100-20, and SLT datasets, and it surpasses the current state-of-the-art methods in neighbor clustering on these datasets. Particularly, AC-VAE achieves 95% and 82% accuracy on CIFAR10 dataset when the average neighbor cluster sizes are 10 and 100, surpassing the current state-of-the-art method by a margin of 10%. Our main innovations and contributions can be summarized as follows:

- This work proposed a VAE based network to encode the image into the representation with its boundary information. The representation and boundary information are retrieved from the multivariate normal distribution, which encoded from the image. The efficacy of the adaptive boundary is demonstrated by neighbor clustering results.

- In this work, a loss function is proposed based on consistency regulation to train the VAE-based network for extracting the high-level semantic feature from the image. Experiments demonstrate that the proposed method assigns close vectors to images with similar semantic meanings.

- This work proposed a clustering method to take advantage of the adaptive boundary of each representation. The proposed method delivers high accuracy neighbor clusters. Besides, the neighbor clusters are found converge within the clustering range ($\alpha \leq 2$), and the self-supervised learning framework utilizing the converged clusters delivers competitive results without the need of a pre-selecting parameter $k$.

## 2 RELATED WORKS

Many frameworks cluster the dataset directly into semantic classes, and train the network in an end-to-end manner (Asano et al., 2019; Caron et al., 2019; Haeusser et al., 2018; Yang et al., 2016; Xie et al., 2016). Although the end-to-end training method is easy to apply, the network's initialization largely influences these frameworks' performance. Therefore, complex mechanisms (such as cluster

reassignment) are needed to assist the clustering process. As an alternative approach, methods (Caron et al., 2018; Hu et al., 2017; Yang et al., 2016) based on maximizing the mutual information between image augmentations are proposed to address this issue.

In contrast to end-to-end training, the multi-stage method (Van Gansbeke et al., 2020) is introduced, which first aim to obtain accurate neighbor clusters from the representation learning, then apply these neighbor clusters in the followed finetuning, and this method made breakthroughs in unsupervised classification. This method depend mainly on the accurate neighbor cluster results from the representation learning. A large number of representation learning methods (Doersch et al., 2015; Gidaris et al., 2018; Noroozi & Favaro, 2016; Pathak et al., 2016; Zhang et al., 2016) have been introduced, and these methods usually assign pretext tasks to the network, and the network learns the image representation by solving these tasks. However, most of these methods aim to use the learned representations to serve the following supervised or semi-supervised tasks. Therefore, the neighbor clustering performance of these learned representations is not optimized, and few of these methods have strong neighbor clustering performance. SimCLR (Chen et al., 2020a) and MoCo (He et al., 2020) utilizing consistency regularization outstanding other methods in neighbor clustering performance and assisted SCAN framework (Van Gansbeke et al., 2020) to reach the state-of-the-art results in unsupervised classification tasks. However, the SCAN framework needs to pre-select the super parameter $k$ to perform the KNN clustering from the starter.

This paper aims to provide an adaptive clustering method that needs no super parameters by creating the boundary for each representation. The representation and its boundary information are retrieved from a VAE based structure. VAE based networks are typically used for image generation tasks (Kingma & Welling, 2013; Razavi et al., 2019) or disentanglement tasks (Higgins et al., 2016; Burgess et al., 2018). Although VAE shows the potential to encode images into multivariate normal distributions (Razavi et al., 2019; Burgess et al., 2018), the efficacy of utilizing VAE to extracting high-level representations is not heavily studied. Besides, VAEs are usually trained by different forms of image recovery tasks, which keep VAE away from extracting high-level semantic features.

This paper adopts the consistency regulation to train the proposed VAE based network for extracting the high-level representation and its boundary information. Moreover, a clustering method is proposed to utilize this boundary information to deliver adaptive cluster results. In the end, these adaptive clusters are utilized in the unsupervised classification task.

## 3 METHOD

The following sections presents the generative network that produces the representation and its boundary information first, and then introduces the cluster method that benefits from the boundary information.

### 3.1 GENERATIVE MODEL

In the unsupervised setup, the ground truth of the desired image representation is not available. However, there are general assumptions about the desired presentation's behavior pattern, i.e., how desired representations should interact with each other, such as images showing the same kind of object should have similar semantic representations. This paper introduces the latent vector that controls the representation's behavior pattern and utilizes a VEA based network to generate representation from this latent vector. The proposed network aims to generate representations that follow the behavior assumption of the expected representation. In that case, the generated representation and the expected one would share the same latent vector, as the latent vector decides the representation's behavior pattern. However, the generated representation may differ from the expected one, even when they have the same behavior pattern. It is hard to directly generate the desired representation, which needs the ground truth to train from the starter. Therefore, this paper adopts the latent vector as a close approximation of the desired representation.

The proposed VAE based network is shown in Figure 2. This paper states the latent vector as a multivariate normal distribution $\mathcal{N}(x; \mu, \Sigma)$ encoded from the image $x$ by an encoder $e(x)$. Then, a sample $z$ is drawn from this distribution with a stochastic process. This random process creates variations that support tryouts to find this latent vector's acceptable range in the training stage. Besides, the distribution's mean $\mu$ is also taken as a standard latent vector to provide a short-cut for

better encoder training. As the encoder is a deep neural network for extracting high-level semantic features, the stochastically sampled latent vector $z$ cannot provide a stable guide to train the deep encoder. The sum of the sampled latent vector $z$ and the standard latent vector $\mu$ is fed into the decoder $d(x)$ to generate the representation $r$.

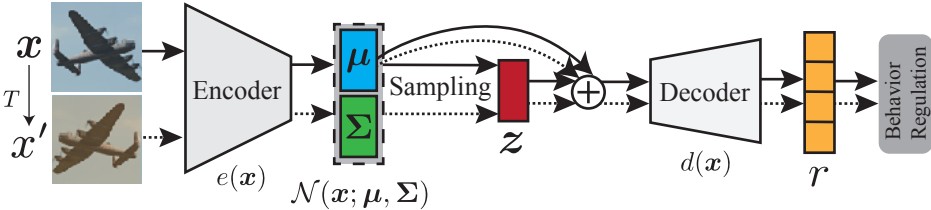

Figure 2: The framework of the proposed VAE based network.

The network is trained by the behavior regulations, which impose regulations on the generated representation. This work adopts consistency regulation, a commonly used behavior assumption of the semantic representation, which regulates an image and its argumentations to have the same semantic representation. The consistency regulation can be performed by minimizing the behavior loss stated in Equation (1).

$$BL_x = d(r_i, r_i') = d(v(x_i), v(T(x_i))),  \tag{1}$$

in which, $T(x_i)$ is the augmentation of image $x_i$, $r_i$ is the representation of $x_i$ generated by the proposed network $v(.)$, and $d(,)$ measures the distance of two representations.

As the proposed network is based on VAE, the loss function of the vanilla VAE (Kingma & Welling, 2013) is adapted to train the proposed network by replacing its image recovery loss with the behavior loss $BL_x$. The loss function is used to train the proposed network is shown in Equation (2).

$$E_{z \sim Q(z|x)}[log(P(r|z)) - KL[Q(z|x)||P(z)],  \tag{2}$$

in which, $P(r|z)$ is distribution of the representation $r$ generate by latent vector $z$, $P(z)$ is the distribution of the latent vector, $Q(z|x)$ is the distribution of $z$ given $x$, and $KL[Q(z|x)||P(z)]$ is the KL divergence between $Q(z|x)$ and $P(z)$.

As mentioned earlier, the latent distribution will act as a close approximation of the desired representation. The mean $\mu$ will be regarded as the image representation, and the z-score of the distribution characterizes its boundary information.

## 3.2 NEIGHBOR CLUSTERING METHOD

This work clusters the images based on the boundary information based on z-score. The insight is that, when an image is encoded into the distribution, the image's close variations should be located within a small z-score range of this distribution. Figure 3 (a) illustrates the representation and its boundary information, z-score range, in a five-dimensional distribution. For neighbor clustering, the neighbor that its means $\mu$ fall into the required z-score ranges will be clustered, and an illustrated cluster criterion is shown in Figure 3 (b). This cluster criterion is strict, as it requires the clustered neighbor not only close to the root of the cluster but also has a similar flow as the root.

For fast calculation, this work proposes a z-score based distance-vector, in which each element of this vector corresponds to the distance at each dimension. The z-score is used because the direct compression of z-scores between different normal distributions is veiled. The proposed z-score based distance-vector $d_{(x_i, x_j)}$ between $x_i$ and $x_j$ shows in Equation (3).

$$d_{(x_i, x_j)} = \alpha \frac{abs(\mu_i - \mu_j)}{2\sigma_i} - 0.5,  \tag{3}$$

in which, $\boldsymbol{\mu}_i$ is the mean of $x_i$'s latent distribution, and $\sigma_i$ is the diagonal elements of the covariance matrix $\boldsymbol{\Sigma}$. The $\alpha$ controls the z-score range to be used. When $\alpha = 1$, the distance is normalized by the z-score range [-1, 1]. This work will expand the z-score range to increase the cluster size until it reaches [-2,2], covering more than 95% of the normal distribution. However, the sample falls out of the z-score range of [-2,2] is most unlikely from this distribution's population, therefore, the z-score range is limited within [-2, 2]. To be clustered, all element of the z-score based distance vector should not larger than 0. Besides, $\boldsymbol{d}_{(x_i,x_j)}$ may not equal $\boldsymbol{d}_{(x_i,x_i)}$, as the z-score range of different representations may differ from each other, as demonstrated in Figure 3 (c).

By modifying $\alpha$ in Equation (3), the z-score based distance will change accordingly. The cluster threshold $\alpha$ indicates how strict the clustering criterion is: small $\alpha$ will tighten the cluster restriction, and large $\alpha$ will reduce the restriction. As the experiment in section 4 will demonstrate, cluster converging after $\alpha$ surpasses a certain value are observed on all evaluated datasets, so the $\alpha$ needs no fine-tuning for each dataset.

Notably, the early converge is observed in the experiments, in which the cluster size stops from increasing before reached the desired cluster number. This situation is introduced by the strict clustering method, which requires the neighbor representation to satisfy the criterion in all dimensions. In some cases, the clustering criteria are hard to reach; hence the clustering process stops. To address this issue, this work introduces the loose match strategy and a parameter $\theta$ ( $0 < \theta < 1$ ). Instant of requiring a full match in every dimension as the standard clustering process, a certain mismatch, $1 - \theta$, is accepted, as demonstrated in Figure 3 (d). The loose match strategy is a backup method and is unnecessary for the unsupervised classification, which will be demonstrated in the experiment section.

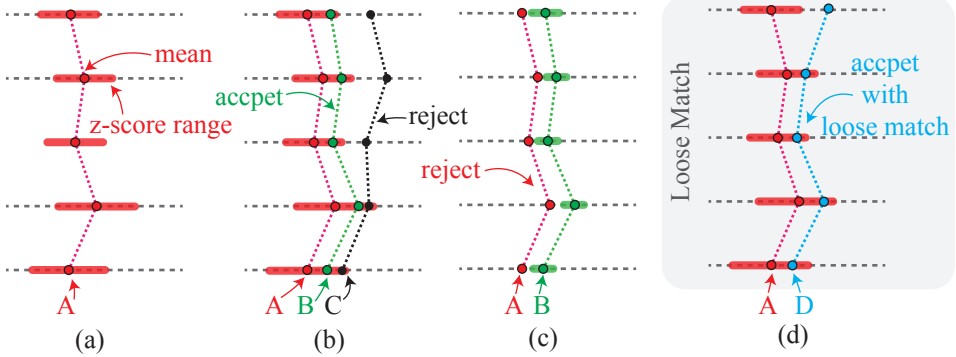

Figure 3: (a) Representation A is illustrated as a five-dimensional distribution. The dots represent the mean of each dimension, and its z-score range is indicated by the bars; (b) Representation B is clustered by A, as all the means of B fall into the z-score range of A. However, representation C cannot join the A's cluster as some of C's means fall out of the required z-score range. (c) Representation A cannot be clustered by B, even representation B is clustered by A. Because B has a narrow z-score range; (d) With the loose match strategy, the representation D can be accepted to join the A's cluster, although one of its means cannot fall into the required z-score range.

## 4 EXPERIMENTS

This paper first evaluates the neighbor cluster performance of the proposed method. Then it uses the neighbor cluster results in a self-supervised classification framework to demonstrate the potential to adapt the proposed method to support self-supervised classification. At last, additional feature of the proposed method is introduced.

### 4.1 EXPERIMENTAL SETUP

Experiments are performed on CIFAR10 (Krizhevsky et al., 2009), CIFAR100-20 (Krizhevsky et al., 2009) and STL10 (Coates et al., 2011). The proposed network is trained on the training set of all

Table 1: Neighbor Cluster Results on Training Set

| Methods | CIFAR10 | | | CIFAR100-20 | | | STL | | |
|---|---|---|---|---|---|---|---|---|---|
| | 10 | 50 | 100 | 10 | 50 | 100 | 10 | 50 | 100 |
| Rot + KNN | 65.58 | 62.06 | 60.08 | 51.76 | 51.41 | 50.38 | 60.69 | 54.71 | 50.02 |
| InstDisc 1 + KNN | 74.73 | 73.96 | 73.08 | 56.66 | 56.41 | 54.28 | 71.52 | 61.82 | 55.38 |
| InstDisc 2 + KNN | 71.33 | 67.78 | 64.38 | 57.76 | 56.36 | 55.68 | 66.24 | 59.71 | 54.78 |
| MoCo + KNN | 75.28 | 70.72 | 68.27 | 56.92 | 56.49 | 54.83 | 71.97 | 57.66 | 59.89 |
| SimCLR + KNN | 81.53 | 76.76 | 73.58 | 61.56 | 54.05 | 49.33 | 75.82 | 67.11 | 62.58 |
| VAE + KNN | 75.71 | 73.34 | 72.32 | 59.27 | 52.04 | 48.38 | 74.71 | 72.83.08 | 58.92 |
| AC-VAE | **95.25** | **87.72** | **82.27** | **68.75** | **65.51** | **62.48** | **92.75** | **87.93** | **85.42** |

datasets. For all experiences, the same set of configurations is applied. A ResNet-34 is adopted as the encoder network, and a two-layer full connection is utilized as the decoder network. The latent distribution dimension is set as 512, and the decoded representation vector has a size of 64. Both the encoder and the decoder networks are initialized randomly. Besides, the NT_Xent loss (Chen et al., 2020a) and its augmentation strategy are used for the behavior loss implementation.

## 4.2 EVALUATION OF NEIGHBOR CLUSTER

The neighbor cluster performance is evaluated under cluster sizes of 10, 50, and 100, as it is desired that a clustering method can maintain high accuracy when keep increasing the cluster size. The accuracy of the neighbor cluster is obtained by averaging all neighbor clusters' accuracy. While in the proposed method, neighbor clusters have different cluster sizes, the average cluster size is used for convenient comparison. For comparison, two instant discrimination based methods (Caron et al., 2018; 2019), Rot (Coates et al., 2011), SimCLR (Doersch et al., 2015), and MoCo (Gidaris et al., 2018) are chosen, as they all use consistency regulation to perform unsupervised representation learning. To get the cluster results, a KNN is applied to the learned representation. For the proposed VAE based network, both KNN and z-score based methods are employed for clustering. The comparing methods are not suitable to use the z-score based clustering method, as it requires the distributional information. The neighbor cluster accuracy on the training set is studied, as the neighbor cluster results on the training set will be utilized to support the self-supervised classification training.

The neighbor cluster accuracy comparison on the training set is shown in Table 1. The proposed VAE based network with KNN is compared with others to demonstrate its ability to project the same meaning images to close representations. With KNN clustering, the proposed method's accuracy is lower than the state-of-the-art, SimCLR. This is because the sampling process introduced by the proposed network creates uncertainty in the representation, which contributes the decline of accuracy. After the KNN is replaced by the z-score based clustering method, the proposed methods (AC-VAE) outperformed all other methods in all cases. Notably, around 10% increases are found when cluster size is 10 on CIFAR10 and STL datasets. This performance comes from the z-score based cluster method, which uses the boundary information to exclude those nearby samples that do not have the same flow shape. Figure 4 is used to demonstrate the efficacy of the proposed clustering method. In Figure 4, the representations from different classes overlap each other in some areas. It is hard for the method that only considers the distance to deliver a highly accurate cluster.

To utilize the neighbor cluster method in unsupervised classification, the parameter $\alpha$'s effect on the cluster size and cluster accuracy is also studied. The results are shown in Figure 5. Clusters have been found naturally converged in the training set of all three datasets within the range of $0 < \alpha \leq 2$. As shown in Figure 5 (a), the cluster size remains the same after the threshold $\alpha$ reaches a certain point. These results benefit from encoding the image as the distribution. For a distribution, most of its population will fall into its z-score range of [-2, 2] ($\alpha = 2$), covering 95% of the distribution population. During the network training, the VAE-based network should push samples with the similar meanings into this high-possibility region. This analysis matches the experiment results, in which the converges happened before the z-score range expands to [-2, 2], as shown in Figure 5.

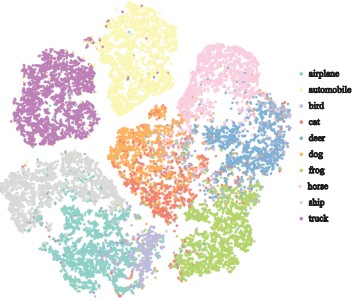

Figure 4: Visualization of representation produced by VAE based network using t-sne (Maaten & Hinton, 2008) for dimensionality reduction.

Table 2: The $\alpha$ and $\theta$ applied in the experiments

| Dataset | CIFAR10 | | | CIFAR100-20 | | | STL | | |
|---|---|---|---|---|---|---|---|---|---|
| Cluster Size | 10 | 50 | 100 | 10 | 50 | 100 | 10 | 50 | 100 |
| $\alpha$ | 0.52 | 0.94 | 1.28 | 1.45 | 1.62 | 1.72 | 0.57 | 1.24 | 1.42 |
| $\theta$ | N/A | N/A | N/A | N/A | 0.95 | 0.91 | N/A | 0.89 | 0.92 |

To get the results mentioned in Table 1, the $\alpha$ and $\theta$ listed in Table 2 are applied. However, these super parameters are precisely selected for easy compassion to the KNN based clustering results. In practice, there is no need to fine-select $\alpha$ nor $\theta$ to reach a specific cluster size unless a particular neighbor cluster size is highly desired.

Figure 5: (a) the clustered size increases as $\alpha$ increases and reaches converged number within $\alpha \leq 2$; (b) the cluster accuracy decreased as the $\alpha$ increases and maintains the accuracy within $\alpha \leq 2$.

## 4.3 SELF SUPERVISED CLASSIFICATION

In order to demonstrate the efficacy of utilizing the proposed method to perform unsupervised classification, the self-supervised classification framework SCAN (Van Gansbeke et al., 2020) is adapted by replacing its KNN based neighbor clustering method with the proposed method to perform the unsupervised classification. The converged clusters showed in section 4.2 are utilized, in which all the clusters are naturally converged without using the loose match strategy nor fine-tuning $\alpha$. Therefore, the self-adaptive cluster results are used to perform the unsupervised classification. As Table 3 demonstrated, the framework utilizing the adaptive cluster results outperform most of the comparing methods. When compared to the state-of-the-art (SCAN, Van Gansbeke et al. (2020)), the proposed method still delivers competitive results without selecting super parameter $k$.

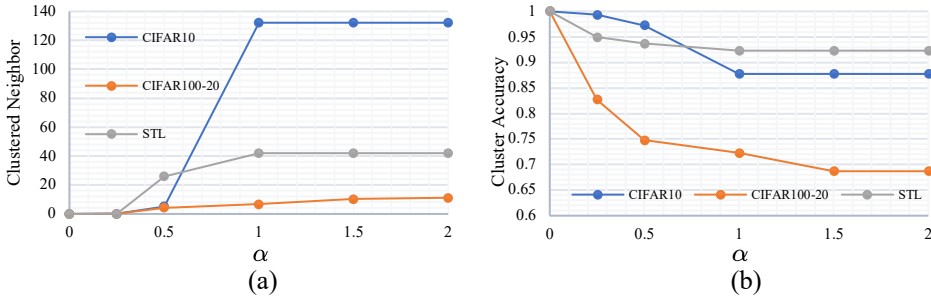

Table 3: The Unsupervised Classfication results

| Methods | CIFAR10 | CIFAR100-20 | STL |
|---|---|---|---|
| DEC (Xie et al., 2016) | 30.1 | 18.5 | 35.9 |
| ADC (Haeusser et al., 2018) | 32.5 | 16.0 | 53.0 |
| DeepCluster (Caron et al., 2018) | 37.4 | 18.9 | 33.4 |
| DAC (Chang et al., 2017) | 52.2 | 23.8 | 47.0 |
| IIC (Ji et al., 2019) | 61.7 | 25.7 | 59.6 |
| SCAN with KNN clustering | 87.6 | 45.9 | 76.7 |
| SCAN with AC-VAE | 79.2 | 40.2 | 71.8 |

## 4.4 OTHER ADVANTAGES

The proposed method also has additional advantages, desired in the unsupervised setup, such as prototype selecting. As different representation has different boundary information, some samples will cluster far more neighbors than others under the same z-score range, so each class's prototype can be easily identified as those who cluster the most neighbors. The selected prototypes of each dataset are shown in Figure 6.

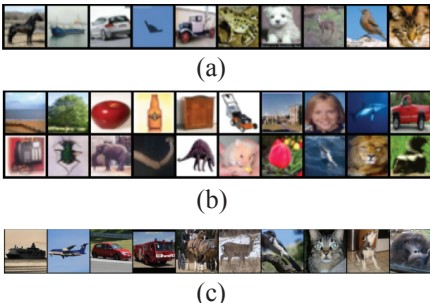

(a)

(b)

(c)

Figure 6: Prototype images on the different datasets: (a) Cifar10, (b) Cifar100-20, (c) STL.

## 5 CONCLUSION

This paper proposes AC-VAE, including a VAE based network and a clustering method. The VAE based network encodes the image into the multivariate normal distributions. The semantic representation and its boundary can be retrieved from this distribution. The clustering method takes advantage of the boundary information to achieve adaptive neighbor clustering. Experiments demonstrate that the proposed VAE-based network has the ability to project images with the same semantic meaning into close representations. Experiments also show the efficacy of the proposed method to from adaptive clusters. This work attempts to push the edge of fully unsupervised classification by omitting a critical super-parameter $k$ in the state-of-the-art method. Experiments show that the naturally converged cluster supports the unsupervised classification framework to deliver competitive results.

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
