# OpenReview forum: "AC-VAE: Learning Semantic Representation with VAE for Adaptive Clustering"
_ICLR.cc/2021/Conference — Reject_

### Official Review · AnonReviewer4 · 2020-10-28
**Promising work towards image clustering may benefit from a more clear presentation and precise terminology.**

**Rating:** 5
**Confidence:** 3

**Review:**

Summary of the paper:

This paper proposes an encoder-decoder architecture for clustering images based on their semantic (high-level) features as well as creating a semantic representation for downstream tasks. The model takes as a basis the encoder part of a variational auto-encoder (VAE) and uses the properties of the distributions induced in the latent space to define cluster boundaries and avoiding the need of setting the parameter K (number of clusters), usually required in methods based on K-Means. In order to induce the encoder to focus on high-level features, a decoder loss based on consistency regulation is defined, which aims at mapping variations of each image, obtained by data augmentation transformations, to similar decoded representation.

Questions and Suggestions:

Shouldn’t be abs(mu_i - x_j) In Equation (3)?

z-scores are usually defined as (mu-x)/std. However, in Equation (3), defining d(x_i, x_j), there is a “-0.5” term. Although this equation is supposed to represent a distance, it can result in negative values when abs(mu_i-mu_j) is small. I suggest removing this term and compare the result against 0.5 instead of 0.0. Also, include the 2 in the denominator in the alpha parameter and make it range in [0.0, 1.0] instead of [-2.0, 2.0].

Equation (2) is not an equation, since it has no equal sign. As it defines a loss function, I suggest using equality to introduce a symbol defining the loss.

Explain Figure 2 in the caption, indicating what the terms mean as well as the doted and solid lines.

Fix typos:
“argumentations” -> “augmentations” (section 3.1)
“Compression” -> “Comparison” (section 3.2)

Pros:

The paper is focusing on important problems in the field of representation learning.
It presents good clustering accuracy results in challenging datasets
It seems to boost self-supervised classification.

Cons:

The presentation and clarity of the article should be better and it would benefit from a review of the English

The use of terminology is not precise, leaving the reader oftentimes confused about what is meant. To give a few examples:
- “Autoencoders”: The architecture does not map inputs to inputs themselves, so strictly speaking it is not an AE model.
- “Adaptive Clustering”: This can mean different things in the literature, for instance, it is also used to mean methods that deal with non-static distributions. But in the article, the term is not clearly defined and just means establishing the number of clusters.

- The use of the words neighbor/neighborhoods: is confusing: “neighbor cluster performance”: is this evaluating the capacity of the model to map similar clusters close to each other (making them neighbors) or of mapping similar samples close to each other  (regular clustering evaluation). Also, “neighbor cluster accuracy”: Is this the same as “Clustering Accuracy”, a standard metric to evaluate clustering methods, or is this evaluating something else? How is it computed?

- Define what these terms mean: “Behaviour Pattern”, “Root of cluster”, “Similar flow as the root”

The review of the literature misses some important articles, which should also be compared with in the experiments:
Vincent Fortuin et al, 2019 “SOM-VAE: Interpretable Discrete Representation Learning on Time Series”
Florent Forest et al 2019 “Deep Embedded SOM: Joint Representation Learning and Self-Organization”

Evaluating Clustering is tricky and the use of only one metric is usually not enough and misleading. I suggest including other metrics such as Normalized Mutual Information and Purity to make the evaluation more sound.

The use of only CIFAR in evaluation is also a problem, as clustering datasets usually present a great variability with different methods. Other papers also consider MNIST and FASHION MNIST, for instance.

The article replaces the parameter K with two other parameters: alpha and theta. Although it states that these parameters are easier to adjust than K, the experiments are not enough to show that, It is missing especially one experiment evaluating the results of other methods with different values of K. It can be the case that with K high enough other methods will produce similar results in downstream tasks.

How many clusters the does method find? Is it the expected number? This is not covered by clustering accuracy.

Conclusion:

Although the paper is interesting and has some novelty aspects, I am not convinced about the soundness of the results due to the lack of evaluation with other metrics and different datasets. Also, I believe that other similar methods, mentioned above, should be included in the comparison. Finally, experiments showing that alpha and theta are easier to adjust than K are needed.

---

### Official Review · AnonReviewer1 · 2020-10-28
**Official Blind Review #1**

**Rating:** 3
**Confidence:** 3

**Review:**

This paper proposes an adaptive neighbor clustering method by estimating normal distribution on the representation space. The proposed neighbor clustering can utilize the acceptable range for each dimension of each instance from the estimated variance, which leads to different size of neighbors for each instance, and improved the neighbor clustering performances. In addition, the proposed neighbor clustering method can replace the KNN-based neighbor clustering in the previous SCAN (Semantic Clustering by Adopting Nearest neighbors) framework for image semantic clustering.

Overall, the paper is hard to follow and understand with many typos, incorrect notations, and especially the main term, VAE. Why can it be called VAE without a decoder which decodes some original input from a compact representation? I think the proposed network contains only the encoder (transforming z to r is also encoding), and it applies the stochastic process in the middle of encoder (512-d z instead of 64-d r). Therefore, it seems that the proposed training can be considered as the previous contrastive learning with stochastic regularization. And, it naturally raises the question that why not imposing the normal distribution on the final representation r? In addition, I do not understand why the sum of the sampled z and \mu is fed into for obtaining r. What if z is solely fed into? What is v in Eq. (1)? Is it d \circ e? How to explicitly connect Eq. (1) and (2)?

In Table 3, The clustering performance drop of SCAN with AC-VAE is not marginal compared to the previous SCAN with KNN, even though the proposed AC-VAE seems to produce significantly improved neighbor clustering results in Table 1. However, there is no analysis on this.

The amortized inference of normal distribution on the (middle of) representation space, and the use of its estimated variance in obtaining reliable neighbors for each instance seems to be make sense, however the motivation, insight, and contribution of the derived framework seem to be very limited.

Typos: SLT -> STL, VEA -> VAE, argumentations -> augmentations, d(x_i, x_i) -> d(x_j, x_i)

---

### Official Review · AnonReviewer5 · 2020-11-06
**Interesting idea, clarifications needed**

**Rating:** 5
**Confidence:** 3

**Review:**

The paper describes the approach  for clustering the image data using z-scores for semantic representation.

While the method seems novel, there are questions to be clarified, and the final rating is dependent on the answers on those questions:

1. The paper states that the method performs adaptive clustering. This statement needs clarification. From that claim I would generally expect that it means adapting to non-stationary data but it has apparently a different meaning in the context of the description of the method. Instead, as the reviewer can see from Figure 3, it is a z-score based cluster assignment. It is unclear what is the precise meaning behind the ‘adaptive’ quality.
2. It is unclear what would happen if the same point is matched to different clusters. Would it be possible to formalise it all as an algorithm so that the described procedure is reproducible?
3. Section 3.2 Besides, $d(x_i,x_j)$ may not be equal to $d(x_i,x_i)$. There is an apparent typo: should be $d(x_j,x_i)$?
4. On section 4.4, while the prototypes are shown, it would also be important to see the examples of neighbours of the prototype by their increased distance.
5. It is said in the discussion of the Eq 1 that *" The consistency regulation can be performed by minimizing the behavior loss stated in Equation (1): $BL_x = d(r_i,r_i^′) = d(v(x_i),v(T(x_i)))$ in which, $T(x_i)$ is the augmentation of image $x_i$, $r_i$ is the representation of $x_i$ generated by the proposed network $v(\cdot)$, and $d(\cdot)$ measures the distance of two representations.”* It is not clear how exactly the augmentation of image is constructed and what distance is used. Could this loss function be described in more detail?

6. For Table 1, the results are shown with specially selected parameters. As the comment goes that it is possible to do it without task-specific $\alpha$ parameter, would it be possible to show these results for the common value of $\alpha$?
7. It is said in section 4.2 that *'With KNN clustering, the proposed method’s accuracy is lower than the state-of-the-art, SimCLR. This is because the sampling process introduced by the proposed network creates uncertainty in the representation, which contributes the decline of accuracy.’* As I understand, the stochastic representation is essential for the z-score (but not for KNN). Therefore, is there a way to support this claim by the ablation study, i.e. produce a deterministic (autoencoder based, for example) representation and see what the accuracy would be with the deterministic representation + KNN ?

---

### Decision · Program_Chairs · 2021-01-07
**Final Decision**

**Decision:**

Reject

**Comment:**

This paper addresses automatically learning the neighborhood size (they call adaptive neighbor support) for unsupervised representation learning with a VAE.  The neighborhood size is determined based on z-scores from by estimating a normal distribution in the latent space.

The paper is poorly written.  There are several grammatical errors and typos that distracts from understanding the paper.  In addition, the use of terminology is not precise, which adds to the confusion, as pointed out by the reviewers.

AC-VAE is better than VAE+KNN in Table 1 but worse in SCAN with KNN in Table 3.  Further analysis to understand why this is so is needed.

Additional measures of cluster quality is recommended.

As pointed out by the reviewers, this paper is below the acceptance threshold for ICLR.  The reviewers provided several constructive suggestions.  Please refer to detailed reviewer comments to help you improve your paper.